# Wild-Type KRAS Allele Effects on Druggable Targets in KRAS Mutant Lung Adenocarcinomas

**DOI:** 10.3390/genes12091402

**Published:** 2021-09-11

**Authors:** Elisa Baldelli, Emna El Gazzah, John Conor Moran, Kimberley A. Hodge, Zarko Manojlovic, Rania Bassiouni, John D. Carpten, Vienna Ludovini, Sara Baglivo, Lucio Crinò, Fortunato Bianconi, Ting Dong, Jeremy Loffredo, Emanuel F. Petricoin, Mariaelena Pierobon

**Affiliations:** 1Center for Applied Proteomics & Molecular Medicine, George Mason University, Manassas, VA 20110, USA; ebaldell@gmu.edu (E.B.); eelgazza@gmu.edu (E.E.G.); jcmoran@med.miami.edu (J.C.M.); khodge5@gmu.edu (K.A.H.); tdong1@gmu.edu (T.D.); loffredoj@whirc.org (J.L.); epetrico@gmu.edu (E.F.P.); 2School of Systems Biology, George Mason University, Manassas, VA 20110, USA; 3Keck School of Medicine, University of Southern California, Los Angeles, CA 90033, USA; zmanojlo@usc.edu (Z.M.); rbassiou@usc.edu (R.B.); carpten@usc.edu (J.D.C.); 4Division of Medical Oncology, S. Maria della Misericordia Hospital, 06156 Perugia, Italy; vienna.ludovini@ospedale.perugia.it (V.L.); baglivosara@gmail.com (S.B.); 5Department of Medical Oncology, Istituto Scientifico Romagnolo per lo Studio e la Cura dei Tumori (IRST) IRCCS, 47014 Meldola, Italy; lucio.crino@irst.emr.it; 6Independent Researcher, Belvedere 44, Montefalco, 06036 Perugia, Italy; fortunato.bianconi@gmail.com

**Keywords:** non-small cell lung cancer, KRAS, zygosity, drug target, reverse-phase protein microarray

## Abstract

*KRAS* mutations are one of the most common oncogenic drivers in non-small cell lung cancer (NSCLC) and in lung adenocarcinomas in particular. Development of therapeutics targeting KRAS has been incredibly challenging, prompting indirect inhibition of downstream targets such as MEK and ERK. Such inhibitors, unfortunately, come with limited clinical efficacy, and therefore the demand for developing novel therapeutic strategies remains an urgent need for these patients. Exploring the influence of wild-type (WT) *KRAS* on druggable targets can uncover new vulnerabilities for the treatment of *KRAS* mutant lung adenocarcinomas. Using commercially available *KRAS* mutant lung adenocarcinoma cell lines, we explored the influence of WT *KRAS* on signaling networks and druggable targets. Expression and/or activation of 183 signaling proteins, most of which are targets of FDA-approved drugs, were captured by reverse-phase protein microarray (RPPA). Selected findings were validated on a cohort of 23 surgical biospecimens using the RPPA. Kinase-driven signatures associated with the presence of the *KRAS* WT allele were detected along the MAPK and AKT/mTOR signaling pathway and alterations of cell cycle regulators. FoxM1 emerged as a potential vulnerability of tumors retaining the *KRAS* WT allele both in cell lines and in the clinical samples. Our findings suggest that loss of WT *KRAS* impacts on signaling events and druggable targets in *KRAS* mutant lung adenocarcinomas.

## 1. Introduction

Mutational activation of the RAS superfamily of small guanosine triphosphate (GTP)-binding proteins is one of the first described and most common oncogenic events in human cancer [1,2]. Of the three ubiquitously expressed RAS proteins, *KRAS* is the most frequently mutated, making up to 85% of all *RAS* mutations, followed by *NRAS* (11%) and *HRAS* (4%). *KRAS* mutations are frequent genomic events in lung cancer, especially in Non-Small Cell Lung Cancers (NSCLC). Within NSCLC, *KRAS* mutations are almost exclusively seen in adenocarcinomas, with up to 40% of adenocarcinoma harboring oncogenic mutations of the *KRAS* gene. These mutations are rarely detected in the other lung cancer subtypes. Across the forty-four oncogenic *KRAS* mutations identified, 90% occur at codon 12, with G12C (40%) and G12V (22%) mutations being the most frequently observed in lung adenocarcinomas collectively [2,3,4].

Undoubtedly, the introduction of precision medicine has revolutionized treatment for lung cancer patients. For example, small kinase inhibitors able to modulate the activity of aberrantly activated epidermal growth factor receptors (EGFRs) are routinely used as first-line treatment options for *EGFR* mutant lung adenocarcinomas [5]. These targeted molecules have significantly improved response to treatment and survival for these patients. However, targeting adenocarcinomas harboring mutations of the EGFR downstream substrate KRAS still remains a significant unmet clinical need in oncology. Historically, developing therapeutics able to modulate KRAS activity has been extremely challenging due to its small size, high affinity for GTP, and the lack of targetable hydrophobic pockets [3]. Direct inhibition of KRAS has been attempted with agents modulating its post-translational modifications, such as farnesyl transferase inhibitors, but has shown limited clinical efficacy [6,7,8,9]. Indirect inhibition of KRAS activity by targeting its downstream substrates, such as members of the MAPK and the PI3K/AKT signaling pathway, has also met limited success in the clinic [10,11,12]. Targeting mutant KRAS remains a “holy grail” in cancer research [3]. Recently, efforts toward developing *KRAS*-mutant-specific inhibitors have opened new therapeutic opportunities for treating *KRAS* mutant NSCLC. According to the ongoing CodeBreak 100 study, treatment with the small G12C inhibitor Sotorasib (or AMG510) in combination with standard chemotherapy has shown objective remission in 33% of enrolled patients [13,14,15]. Given these encouraging results, the FDA granted AMG510 approval as a treatment option for locally advanced or metastatic NSCLCs with G12C *KRAS* mutations. Another promising G12C inhibitor, Adagrasib, is also showing clinical activity in patients with advanced and previously treated NSCLC. According to the KRYSTAL-1 phase 1/2 trial results, in patients with advanced NSCLC G12C *KRAS* mutant tumors previously treated with chemotherapy and anti-PDL1 therapy, Adagrasib yielded a 96% disease control rate with a 45% objective response rate [16]. However, despite such promising data, maximizing the effect of these inhibitors as a monotherapy and understanding mechanisms of acquired resistance are still under investigation [13,14,15,16]. Thus, the advancement of existing options and development of novel therapeutic strategies for *KRAS*-mutated cancers heavily relies on further understanding the context in which *KRAS* mutant signaling occurs.

While most research efforts have been directed toward dissecting specific *KRAS* point mutations, the role of wild-type (WT) KRAS in the carcinogenesis of mutant tumors has only been partially explored. Previous research has indicated that the WT KRAS protein can either promote or inhibit tumor progression in KRAS mutant tumors in a context-dependent manner [17]. In fact, it is now appreciated that the competitive fitness of *KRAS* mutant cancer cells is achieved through continuously tuning the mutant and WT allele product in a context-specific manner. From a therapeutic perspective, previous studies focusing on *KRAS* mutant colorectal cancer and acute myeloid leukemia have indicated that the WT copy of the *KRAS* gene is associated with resistance to treatment with MEK inhibition [17,18]. However, the role of the WT allele on the overall signaling network of these cancer cells is still not fully understood. To fill this gap, this study explored activation of signaling transduction networks in *KRAS* mutant lung adenocarcinomas lacking or retaining the WT copy of the *KRAS* gene. Such approach provided us the unique opportunity to capture the effects of KRAS homo- and heterozygous mutations on druggable signal transduction molecules. Exploring the impact different genotypes can have on cell signaling events will help identify distinct druggable vulnerabilities and potentially advance precision medicine for *KRAS* mutant lung adenocarcinoma patients.

## 2. Materials and Methods

### 2.1. NSCLC Cell Lines and Cultures

A panel of eleven human adenocarcinoma cell lines including the A427, A549, Calu-3, H1373, H1734, H1838, H2122, H23, H358, H522 and SK-LU-1 models was obtained from the American Type Culture Collection (ATCC, Manassas, VA, USA). Following manufacturer’s instructions, cell cultures were maintained in media (F-12K for the A549; Eagle’s MEM for the A427, Calu-3, and SK-LU-1; and RPMI-1640 for all the remaining cell lines) supplemented with 10% fetal bovine serum (ATCC, Manassas, VA, USA) in a humidified atmosphere with 5% CO_2_ at 37 °C. Cell lines were sub-cultured using trypsin/EDTA (ATCC, Manassas, VA, USA) at a ratio ranging between 1:3 and 1:6 based on cell lines’ proliferation rates. Based on the COSMIC database (COSMIC, Cell Lines Project, https://cancer.sanger.ac.uk/cell_lines, accessed on 4 September 2018), the A549, H2122 and H1373 cell lines harbor *KRAS* homozygous mutations; the A427, H1734, H23, H358 and SK-LU-1 have heterozygous *KRAS* mutations; and the Calu-3, H1838 and H522 were established from *KRAS* WT tumors. Mutational status of mutant models was confirmed by polymerase chain reaction (PCR). 

### 2.2. Cell Viability Assay

Cell viability assay was performed using CellTiter-Glo Luminescent Cell Viability Assay (Promega, Madison, WI, USA) following manufacturer’s instructions. In brief, for each cell line, a cell suspension containing 5000 to 7500 viable cells was seeded in 96-well plates 24 h before treatment with either the small kinase MEK inhibitor Selumetinib (AZD6244, Selleckchem, Houston, TX, USA), the Cdk 4/6 inhibitor Palbociclib (Selleckchem, Houston, TX, USA) or the thiazole antibiotic known to modulate FoxM1 expression and transcriptional activity Siomycin (Cayman Chemical, Ann Arbor, MI, USA). Number of plated cells was selected based on each line’s proliferation rate. Compounds were dissolved in dimethyl sulfoxide (DMSO) (Selumetinib and Siomycin) or phosphate-buffered saline (PBS) (Palbociclib), and cells were treated in a 2-fold serial dilution curve ranging from 0.15 μM to 150 μM for Selumetinib, from 1.12 μM to 10 μM for Palbociclib and from 0.25 μM to 1.25 μM for Siomycin. Matched PBS and DMSO control data were collected for each dilution point. Independent biological replicates (*n* = 4) were collected for each cell line. After 72 h of incubation with the compounds, plates were brought to room temperature for 30 min. Media were replaced with a 1:1 solution of CellTiter-Glo and fresh media, and cells were lysed on an orbital shaker at room temperature for 5 min. Luminescence signal was measured using a Beckman Coulter DTX 880 microplate reader (Beckman Coulter, Brea, CA, USA) [19].

### 2.3. Cell Lysate Preparation for Signaling Network Analysis

Cell lines profiled for signaling transduction analysis were seeded in technical replicates (*n* = 6) in 6-well plates and cultured until 80% confluent. Cells were washed twice with PBS (Invitrogen Life Technologies, Carlsbad, CA, USA) and lysed in Tissue Protein Extraction Reagent (T-PER) (Thermo Fisher Scientific, Waltham, MA, USA) supplemented with 300 mM sodium chloride and a mixture of protease and phosphatase inhibitors to prevent protein degradation and preserve the integrity of the phosphoproteome, as previously described [19]. Protein concentration in each sample was assessed using the Coomassie Protein Assay Kit (Thermo Fisher Scientific, Waltham, MA, USA) following manufacturer’s instructions. Lysates were first diluted to 1 µg/µL in T-PER and subsequently brought to a final concentration of 0.5 µg/µL in 2X Tris-Glycine SDS Sample buffer (Invitrogen Life Technologies, Carlsbad, CA, USA) supplemented with 5% β-mercaptoethanol (Sigma-Aldrich, St. Louis, MO, USA). Lastly, lysates were boiled for 8 min at 100 °C and stored at −80 °C.

### 2.4. NSCLC Tissue Collection

A total of 23 retrospective biospecimens obtained from surgically treated *KRAS* mutant/*EGFR* WT lung adenocarcinomas were collected between 2009 and 2013 at the S. Maria della Misericordia Hospital (Perugia, Italy). The local Institutional Review Board approved the protocol for this study, and written voluntary informed consent was collected from each patient before surgical removal of the tumor. Specimens were snap-frozen within 30 min upon surgical resection, embedded in optimal cutting temperature compound (OCT) and stored at −80 °C. *EGFR* (Exons 18–21) and *KRAS* (Exons 1–2) mutation status was analyzed by the enrolling institution using Sanger sequencing as previously described [20].

### 2.5. Laser Capture Microdissection

For each biospecimen, pure tumor epithelia (>95%) were isolated for downstream DNA and protein analysis using laser capture microdissection (LCM). Each sample was cut into 8μm sections, mounted on uncharged glass slides, and stored at −80 °C until microdissected. A representative section stained with Hematoxylin (Sigma Aldrich, St. Louis, MO, USA) and Eosin (Sigma Aldrich, St. Louis, MO, USA) was evaluated to confirm the presence and amount of tumor in each specimen. LCM-dedicated slides were stained with the HistoGene LCM Frozen Section Staining Kit (Life Technologies, Carlsbad, CA, USA) for PCR analysis and as previously described for RPPA analysis [21,22]. Isolation of pure tumor epithelium from the surrounding microenvironment was performed using a PixCell IR microdissection system (Arcturus Bioscience, Mountain View, CA, USA), and isolated cells were collected on a single CapSure Macro LCM cap (Arcturus Bioscience, Mountain View, CA, USA). Approximately 2000 tumor cells were collected for DNA analysis and nearly 3000 cells were collected for the RPPA assay. RPPA designated samples were lysed in a 1:1 solution of 2X Tris-Glycine SDS Sample buffer (Invitrogen Life Technologies, Carlsbad, CA, USA) and T-PER (Thermo Fisher Scientific Waltham, MA, USA) supplemented with 2.5% of β-mercaptoethanol (Sigma, St. Louis, MO, USA) and store at −80 °C.

### 2.6. DNA Extraction and PCR Amplification

DNA extraction was performed using the Arcturus PicoPure DNA Extraction Kit (Life Technologies, Carlsbad, CA, USA) following manufacturing instructions. In brief, ten microliters of buffer were added to each cap, and samples were incubated at 65 °C for 3 h followed by 10 min at 95 °C. DNA concentration was measured using the Qubit fluorometer (Invitrogen, Waltham, MA, USA); an average of 40 ng/uL of DNA was recovered for each sample. The genomic DNA was subjected to PCR amplification using GoTag Master Mix (Promega, Madison, WI, USA) and the following primers:Forward: 5′-GAGTCTTGCTCTATCGCCAGG-3′
Reverse: 5′-CCTCATCTGCTTGGGATGGAAG-3′

The annealing temperature was adjusted at 60 °C, targeting 35 cycles to generate sufficient amplicon. Sanger sequencing of the amplicons in the region spanning codons 12 and 13 was performed by GENEWIZ, Inc (South Plainfield, NJ, USA), using the forward primer sequence 5′-CGATACACGTCTGCAGTCAACT-3′. For samples returning a high background signal, additional Sanger sequencing was performed using the reverse primer sequence 5′-CCTCATCTGCTTGGGATGGAAG-3′ to ensure appropriate interpretation of the results.

### 2.7. Reverse-Phase Protein Microarray

Cell lysates and microdissected material were immobilized in technical triplicates onto nitrocellulose-coated glass slides (Grace Bio-Labs, Bend, OR, USA) using an Aushon 2470 arrayer (Quanterix, Billerica, MA, USA) equipped with 185 μm pins in two independent sets of arrays. Reference standards were printed along with the experimental samples for internal quality control. Selected arrays were stained with Sypro Ruby Protein Blot Stain (Molecular Probes, Eugene, OR, USA) following the manufacturer’s directions to quantify the total protein amount of each sample and used for normalization. Each array was probed with one polyclonal or monoclonal primary antibody targeting a protein of interest using an automated system (Dako Cytomation, Carpinteria, CA, USA). Primary antibodies were validated for their specificity to the target protein by Western blotting, as previously described [23]. Arrays were probed with a total of 183 antibodies, mainly capturing expression and activation levels of targets and downstream substrates of FDA-approved anti-cancer drugs or investigational agents (Appendix A). Biotinylated anti-rabbit (Vector Laboratories, Inc., Burlingame, CA, USA) or anti-mouse secondary antibodies (Dako Cytomation, Carpinteria, CA, USA) were used to detect the primary antibodies. Signal was detected using a commercially available tyramide-based avidin/biotin amplification system (Catalyzed Signal Amplification System (CSA); Dako Cytomation, Carpinteria, CA, USA) coupled with a fluorescent streptavidin-conjugated IRDye680 dye (LI-COR Biosciences, Lincoln, NE, USA). To capture background and unspecific signal, selected arrays were incubated with the secondary antibodies alone. Images were acquired using a laser scanner (Tecan PowerScanner, Mönnedorf, Switzerland). Spot intensity was quantified using MicroVigene software V5.1.0.0 (VigeneTech, Carlisle, MA, USA) as previously described [19,22]. In brief, background noise and unspecific signal were subtracted from each sample, and intensity values were normalized to the total amount of protein. Lastly, a single RPPA intensity value was generated for each sample by averaging the technical replicates (*n* = 3).

### 2.8. Clonogenic Assay

Clonogenic assay was performed using soft agar colony formation method as previously described [24]. In brief, noble agar was dissolved in de-ionized water, and six-well plates were coated with a 1:1 mixture of fresh media and 1% agar solution. Agar was allowed to solidify for 30 min at room temperature. A549, H2122, H23 and H358 cell suspensions at the concentration of 5000 cell/mL were mixed with a 0.6% agar solution in a 1:1 ratio and seeded on the coated 6-well plates. Cell/agar mixture was allowed to solidify at room temperature for 30 min and was subsequently incubated at 37 °C. A layer of fresh media was maintained over the cells to prevent desiccation. Cells were monitored for colony formation for 21 days, and images were acquired using an IX51 microscope equipped with an Olympus DP72 camera (Shinjuku City, Tokyo, Japan).

### 2.9. Statistical Analysis

To examine frequency and KRAS zygosity in commercially available cell lines across tumor types, KRAS mutation status for all available cell lines was retrieved from the Catalogue of Somatic Mutations in Cancer database v94 (COSMIC, Cell Lines Project, https://cancer.sanger.ac.uk/cell_lines, accessed on 4 September 2018) and the NCI RAS initiative website (https://www.cancer.gov/research/key-initiatives/ras, accessed on 16 June 2021) and cell lines were filtered for KRAS mutation status.

For drug sensitivity testing performed in-house, IC50s were calculated using a linear regression curve fit method after normalization on the vehicle controls using GraphPad v9. IC50 were also retrieved for four MEK and ERK inhibitors, namely Trametinib, ERK-6604, ERK-2440 and Ulixertinib, from the Genomics of Drug Sensitivity in Cancer (GDSC) database (https://www.cancerrxgene.org, accessed on 14 June 2021).

Spearman rank-order correlation coefficients of RPPA-based continuous values were calculated in JMP v5.1 (SAS Institute Inc., Cary, NC, USA) across the 183 measured analytes. Interactions with a correlation coefficient greater than 0.90 were displayed using network maps generated in Gephi v0.9.2. Non-parametric tests were computed to compare expression/activation of the 183 signaling molecules between *KRAS* mutant lines retaining (*KRASm/WT+*) or lacking (*KRASm/WT−*) the WT alleles and between *KRASm/WT+, KRASm/WT−*, and WT cell lines. *p* values < 0.05 were considered significant. Significant findings were displayed using bar graphs created in GraphPad v.6.07. Within each cell line, RPPA values were normalized to baseline values; bar graphs visualize mean and standard error of each group.

## 3. Results

### 3.1. Response to Treatment in KRAS Mutant NSCLC Is Affected by the Presence of the WT KRAS Allele

While retention of the *KRAS* WT allele in *KRAS* mutant tumors has emerged as a factor that modulates resistance to MEK and ERK inhibition in preclinical studies [5,17,25], little is known about the distribution of the WT *KRAS* allele in lung cancers and its clinical implications. To answer this question, we first explored the frequency of *KRAS* WT allele in 116 *KRAS* mutant cell lines profiled by the COSMIC database and by the NCI-funded RAS initiative. *KRAS* mutant tumors lacking the WT allele (*KRASm/WT−*) were more prevalent in lung cancer cell lines compared to other tumor types. Indeed, of the 36 lung cancer models retrieved from the COSMIC database, 47.2% were *KRASm/WT−*, compared to 30.3% in the pancreatic group and 20% in the large intestine lines (Figure 1). WT copy of the *KRAS* gene was frequently retained in lines harboring G12D mutations, regardless of the tumor type. As expected, G12C and G12V mutations were highly prevalent in lung cancers (41.6% and 27.8%, respectively), and a relatively large number of these cell lines (53.3% and 70.0%, respectively) did not retain a copy of the WT allele compared to other codon-specific alterations.

We then sought to evaluate whether response to Selumetinib in lung cancer cell lines harboring *KRAS* oncogenic mutations is affected by the presence of the WT allele as previously described for different tumor types [18,25]. Response to Selumetinib was initially assessed on six commercially available *KRAS* mutant adenocarcinoma cell lines, of which four (H23, H358, H1734 and SK-LU-1) retained a copy of the WT allele. Presence of the WT allele in these models was first retrieved from the COSMIC database and subsequently confirmed by PCR analysis. Cell lines were treated with Selumetinib in an 11-point dilution curve with concentrations ranging from 0.15 μM to 150 μM. As shown in Figure 2, Panel A, cell lines lacking the WT *KRAS* allele were more sensitive to the compound compared to *KRASm/WT+* and to *KRAS* WT cells. Specifically, IC50 ranged between 1.5 and 6.3 μM for *KRASm/WT−* cells compared to 11.3–71 μM for *KRASm/WT+* and 30–390 μM for the *KRAS* WT models. Based on the Genomics of Drug Sensitivity in Cancer (GDSC) data, similar trends were also observed, for the same *KRAS* mutant cell lines, with the MEK and ERK inhibitors Trametinib, ERK-6604, ERK-2440 and Ulixertinib (Figure 2). Taken together, our data suggest that the loss of the WT allele in *KRAS* mutant lung adenocarcinomas is often associated with G12C and G12V mutations, and the loss of the WT copy of the *KRAS* gene modulates response to treatment with compounds targeting KRAS downstream substrates.

### 3.2. KRAS WT Alleles Drive Specific Signaling Alterations in Lung Adenocarcinomas Harboring KRAS Oncogenic Mutations

To dissect kinase-driven signaling events associated with the WT *KRAS* allele in mutant lung adenocarcinomas, we then captured, on a panel of 5 adenocaricnoma lines (A549, H1734, H23, H2122, H358), the expression and activation level of 183 signaling proteins by reverse-phase protein microarray (RPPA). The antibody panel selected for this analysis mainly captured expression or activation of targets and downstream substrates of FDA-approved anti-cancer drugs or investigational agents. To increase the biological relevance of our findings, we focused on changes occurring at the pathway level more than variations of individual proteins.

We first looked at interconnected proteins within the *KRASm/WT+* and *KRASm/WT−* cell lines. Pair-wise correlation coefficients were calculated for each protein, and interconnections with Spearman Rho correlation coefficients greater than 0.90 were used to generate network maps where the dimension of each node is proportional to the number of interconnections of each protein. Spearman Rho correlation coefficients were greater than 0.90 for 354 and 767 interconnections in the *KRASm/WT+* and *KRASm/WT−* cell lines, respectively (Figure 3). Signaling molecules were highly interconnected in the *KRASm/WT−* cell lines, where proteins were mainly grouped in three major clusters (red, yellow and blue). On the contrary, *KRASm/WT+* had overall more sparse interconnections with two main subgroups of proteins clustering together (green and turquoise clusters). Of interest, the Ras protein-specific guanine nucleotide-releasing factor 1 (Ras-GRF1), a regulator of RAS activity, emerged as a central node in both networks. However, in the *KRASm/WT+* lines, the network including Ras-GRF1 S916 also contained a number of receptor tyrosine kinases and downstream signaling molecules such as EGFR Y1173, c-Met, Her2, IRS-1 S612 and B-Raf S445, along with different members of the PKC family, including PKCδ T505, PKCα S652 and PKCα/BII T638/641. In the *KRASm/WT-* cell lines, on the other hand, Ras-GRF1 was interconnected with the cell cycle regulators Ki67, Rb S780, p27Kip, pan-methylated Histone H3 proteins and a number of apoptotic molecules including cleaved Caspases 6, 7 and 9 and BAD S155 (Figure 3).

To further characterize signaling events associated with the *KRAS* WT allele in our models, we then compared the expression or activation of signaling molecules in *KRASm/WT+* and *KRASm/WT−* cells (Figure 4 and Appendix A). Expression of Ras-GRF1 was significantly higher in the *KRASm/WT−* cells compared to *KRASm/WT+* as well as to WT cell lines (Figure 4C and Appendix A). Along with Ras-GRF1, activation of upstream signaling molecules such as EGFR Y1148 and the adaptor protein Shc Y317 was also greater in the *KRASm/WT−* models. The *KRASm/WT−* cells also presented with increased activation of signaling molecules involved in anchorage-independent growth including members of the Src family, FAK Y576/577, CrkL Y207, and Paxillin Y118 (Figure 5A). To explore whether increased activation of adhesion molecules translates into anchorage-independent growth in *KRASm/WT−* cells, we performed a clonogenic assay. A total of 5,000 cells were seeded in soft agar and the number and colony dimensions were compared between *KRASm/WT+* and *KRASm/WT−* lines after 21 days. As expected, *KRASm/WT−* cells formed larger and more abundant colonies compared to the *KRASm/WT+* (Figure 5B).

*KRASm/WT+* cells presented with higher activation of several members of the MAPK pathway compared to *KRASm/WT-* models. For example, Raf activity, measured as phosphorylation levels of B-Raf S445, C-Raf S338 and MEK 1/2 S217/221, was greater in *KRASm/WT+* cell lines compared to *KRASm/WT−* cell lines (Figure 4B). These phosphorylation levels were similar to the one detected in *KRAS* WT lines. ERK 1/2 activation was also enhanced in WT and *KRASm/WT+* cell lines compared to *KRASm/WT−* along with increased phosphorylation on the downstream transcription factor Elk-1 (Figure 4B). Of interest, ERK 1/2 phosphorylation was not different between *KRASm/WT+* and cell lines not harboring a *KRAS* mutation. Similarly, phosphorylation levels of members of the AKT/mTOR signaling axis were higher in *KRASm/WT+* models compared to *KRASm/WT−* cells including mTOR S2448, the downstream substrates p70S6 kinase phosphorylated on residues T389 and T412 and the transcription factors FoxO1 S256 and FoxO1 T24/FoxO3 T32 (Figure 4A). Of interest, while phosphorylation is the direct read-out of kinase activity, many phosphorylation events are also indicators of a protein subcellular localization (e.g., nuclear entry and export, cytoplasmic sequestration, mitochondrial export, etc.). Cytosolic sequestration via phosphorylation events of proteins involved in apoptosis (e.g., phosphorylated BAD, FoxO1 or FoxO3 and YAP) and cytoskeleton organization (pCofilin) were also unique characteristics of the *KRASm/WT+* lines (Appendix A).

Taken together, these data indicate that the presence of a *KRAS* WT allele strongly affects signal transduction events in *KRAS* mutant NSCLC cell lines. *KRA*S WT and *KRASm/WT+* cells present with increased activation of the MAPK and AKT/mTOR signaling axis compared to the *KRASm/WT−* lines. In addition, the presence of a *KRAS* WT allele may modulate downstream signaling activity through aberrant cytosol sequestration of apoptotic signaling molecules.

### 3.3. KRAS Mutant Lung Adenocarcinoma Cells Retaining the WT KRAS Allele Have Aberrant Activation of Cell Cycle Regulators and DNA Repair Mechanisms

While changes in the activation of the MAPK and AKT/mTOR pathways have already been tested as potential therapeutic targets for *KRAS* mutant lesions, in *KRASm/WT−* models we identified distinctive alterations in proteins involved in cell cycle progression compared to *KRASm/WT+* cell lines (Figure 4A). For example, phosphorylation of proteins involved in checkpoint-induced cell cycle arrest and DNA double-strand break repair including ATM S1981 and the downstream substrate Chk1 S345 were increased in *KRASm/WT−* lines. On the contrary, proteins involved in cell cycle progression were overexpressed/activated in the *KRASm/WT+*. A few examples include increased expression of Cyclin A2 and B1, deactivation of the tumor suppressor Rb through phosphorylation of the S780 residue and expression and activation of the transcription factor FoxM1 (Figure 4C). Of interest, when FoxM1 T600 levels were compared between *KRAS* mutant and WT cell lines, its activation was not significantly different between the WT and *KRASm/WT+* lines (Figure 4C).

Because FoxM1 and Rb, two downstream targets of Cdk4/6 activity, were hyperphosphorylated in the *KRASm/WT+* cells, we examined if the Cdk4/6 inhibitor Palbociclib affects proliferation of *KRASm/WT+* and *KRASm/WT−* cell lines differently. Incubation with Palbociclib had a very minimal effect across all cell lines (Figure 6A). We then tested the effect of Siomycin, a thiazole antibiotic known to modulate FoxM1 expression and transcriptional activity, on *KRASm/WT+* and *KRASm/WT−* lines (Figure 6B). Incubation with Siomycin at a 0.6 μM concentration for 72 h reduced cell viability by more than 50% in all *KRASm/WT+* lines (Figure 6C). The *KRASm/WT−* models were minimally affected by the compound and response to Siomycin was independent from the *TP53* mutational status. As expected, incubation with Siomycin reduced activation of FoxM1 in sensitive models. Taken together these data indicate the FoxM1 and its transcriptional activity may represent a therapeutic target for *KRASm/WT+* lung adenocarcinomas.

### 3.4. FoxM1 Activation Is Increased in Surgical Specimens of NSCLC Patients Harboring KRASm/WT+ Lesions

To test the clinical significance of FoxM1 activation, we analyzed a cohort of *KRAS* mutant adenocarcinoma samples collected from patients undergoing surgical resection. Pure tumor epithelia were isolated from the surrounding tumor microenvironment using LCM, and the presence/absence of the WT copy of the *KRAS* gene was determined by PCR (Figure 7). Of the 23 samples analyzed, 21 had retained the WT *KRAS* allele; FoxM1 activation levels were higher in the *KRASm/WT+* tumors compared to the *KRASm/WT−* lesions (Figure 4C). Taken together, these data suggest that activation of FoxM1 is increased in *KRAS* mutant lung adenocarcinomas retaining the WT *KRAS* allele and that downregulation of its transcriptional activity may represent a therapeutic target for this group of patients.

## 4. Discussion

In this study, we used a translational approach to characterize signal transduction events in homo- and heterozygous *KRAS* mutant lung adenocarcinomas with a focus on druggable targets and downstream substrates of FDA-approved or investigational agents. In line with previous findings, our study suggests that the presence of the WT *KRAS* allele significantly alters the biology and potential response to treatment for *KRAS* mutant tumors [18]. From a genetic perspective, data retrieved from the COSMIC database suggest that *KRAS* mutant lung cancer models lack the WT copy of the gene more frequently than other tumor types and such homozygous alterations are mostly seen in cell lines harboring G12C and G12V mutations. Interestingly, these mutations with the least percentage of cell lines retaining a WT allele are frequently associated with a worse prognosis [26]. However, a sub-analysis of the SELECT trial, a clinical study testing the efficacy of Selumetinib plus docetaxel in *KRAS* mutant NSCLC, has suggested a slight increase in progression-free and overall survival in patients with G12V and G12C *KRAS* mutant tumors compared to those harboring other *KRAS* oncogenic mutations [11,27]. Although speculative, as G12C and G12V *KRAS* mutant tumors are more frequently associated with the loss of the WT *KRAS* allele, the increased sensitivity to Selumetinib in combination with docetaxel observed clinically may be due to the absence, in a number of these lesions, of the WT allele of the *KRAS* gene.

As previously reported, our data confirmed a significant difference in response to MEK 1/2 and ERK 1/2 inhibitors in *KRASm/WT−* and *KRASm/WT+* cell lines [18]. Specifically, *KRASm/WT+* are overall less sensitive to agents targeting KRAS downstream substrates. Similar trends were also observed in a study by Ambrogio et al., in which inhibition of downstream KRAS activity with Selumetinib, Trametinib and the dual MEK/CRAF inhibitor CH5126766 was only seen in the absence of a WT *KRAS* [25]. According to this work, heterozygosity modulates mutant KRAS downstream activity through dimerization. Indeed, it is speculated by the authors that dimerization between the mutant and WT KRAS impairs its binding to downstream substrates and ultimately blocks the propagation of mitogenic signals. Accordingly, as previously hypothesized, WT KRAS stabilizes the output of MAPK signaling by buffering out any perturbations [25]. In our analysis, cell lines lacking the *KRAS* WT allele presented with decreased activation of several members of the MAPK pathway, not only compared to the heterozygous lines but also to models without oncogenic *KRAS* mutations. While this observation somewhat contradicts the “buffering” hypothesis, our data confirmed a higher degree of interconnections and cross-talks in *KRASm/WT−* cell lines compared to *KRASm/WT+* models suggesting the loss of the *KRAS* WT allele has profound implications on the signaling network of these tumors.

Because KRAS is a key regulator of many signaling molecules, we next compared signal transduction events across our models. From a signaling perspective, *KRASm/WT−* presented with decreased activation of several members of the MAPK pathway, not only compared to the heterozygous lines, but also to models without oncogenic *KRAS* mutations. Downregulation of MAPK signaling in *KRASm/WT−* was detected along with increased activation of several molecules involved in anchorage-independent growth, suggesting some protection against anoikis is gained when the WT *KRAS* allele is lost. Thus, loss of the WT allele in *KRAS* mutant lung adenocarcinomas and its effect on tumor invasiveness and metastasization should be evaluated in future investigations.

From a therapeutic perspective, while changes in activation of the MAPK and AKT/mTOR pathways were expected and have already been tested as potential therapeutic targets for *KRAS* mutant lesions, cell lines retaining a *KRAS* WT allele presented with distinct expression and activation of proteins involved in the regulation of the cell cycle [17,18,25]. Namely, we observed greater levels of expression of Cyclins A2 and B1, two major regulators of G2/M progression, along with downregulation of proteins involved in earlier phases of the cell cycle such as Cyclin D1 and p27/Kip1. Furthermore, we detected prominent upregulation of expression and activation of the transcription factor FoxM1 and the tumor suppressor Rb uniquely in the *KRAS WT+* cell lines. While activation of FoxM1 and deactivation of Rb in our model appeared to be independent from the G1/S regulators Cdk4/6, it is well known that the expression of the transcription factor FoxM1 increases throughout the cell cycle and regulates expression of key proteins involved in its progression. For example, numerous reports have previously indicated that knockdown of FoxM1 is associated with a marked decrease in Cyclin A and B activity, suggesting a prominent role of FoxM1 in modulating the expression of proteins involved in G2/M transition, as also confirmed by our data [28,29,30,31,32,33]. Moreover, as the MAPK signaling pathway is known to stimulate nuclear translocation of FoxM1 through ERK 1/2 phosphorylation [33,34], the combined elevation of MAPK signaling and FoxM1 in KRAS WT+ cell lines we detected is anticipated.

Prompted by the changes in FoxM1 expression and activity between *KRAS* mutant tumors lacking or retaining the WT allele, we sought to test the effect of Siomycin, a thiazole antibiotic known to inhibit FoxM1 mRNA and protein expression [35], on *KRASm/WT+* and *KRASm/WT−* models. As expected, because expression and activation of FoxM1 were significantly different across comparison groups, inhibition of this node led to a significant reduction in cell viability in the *KRASm/WT+* lines, while it only marginally affected *KRASm/WT−* cells. To validate the clinical relevance of our findings, FoxM1 activation in *KRASm/WT+* was also confirmed in surgical biospecimens collected from NSCLC patients affected by lung adenocarcinoma.

While assessing the role of *KRAS* zygosity can easily be explored in pre-clinical in vitro models and biochemical assays, validating these findings in animal models and clinical samples can be challenging from a technical perspective. The heterogeneous composition of the tissue, where cancer cells are commingled with non-malignant cells, renders capturing zygosity of oncogenic mutations in whole-tissue-based analysis unattainable [36,37]. To overcome these technical limitations and validate the clinical relevance of our observations, we performed our molecular analyses on enriched tumor epithelia isolated via LCM. Thus, the approach proposed may open new opportunities for understanding phenotypic manifestations associated with zygosity, allelic imbalance, and gene dosage effect of oncogenic mutations in human cancers.

Although the identification of druggable genomic alternations has revolutionized cancer treatment and management, predicting response to treatment still remains an unmet need in oncology. Zygosity and the presence of a wild-type copy of a mutant oncogene, in particular, are rarely accounted for in the therapeutic decision-making process for cancer patients. However, emerging data suggest that *KRAS* zygosity may profoundly affect cancers’ phenotypes and responses to targeted compounds. For example, a recent paper by Liu and colleagues suggested that loss of the WT copy of the *KRAS* allele is not a rare event in lung adenocarcinomas and is associated with shorter survival [38]. In addition, this work confirms the unique role of zygosity as an important modulator of response to treatment in patients affected by *KRAS* mutant lung adenocarcinomas. These observations have important clinical implications as they suggest that *KRAS* zygosity may represent an underestimated pathological marker of *KRAS* mutant lung cancers. Understanding the roles and effects of zygosity on signal transduction networks may open new opportunities for allocating patients to targeted treatments, understating mechanisms of resistance, and devising new single agent and combination treatments.

While our data support the hypothesis that retention of the WT copy of the *KRAS* gene in mutant lung adenocarcinomas modulates signaling events and response to treatment, a few study limitations must be addressed. First, our observations are merely descriptive and based on a limited number of cell lines generated from lung adenocarcinomas. Given the highly variable genetic background of these models and different degree of dependency on mutant *KRAS*, generalizing our observations is premature and needs to be validated on larger datasets. Second, because we used cell lines harboring diverse *KRAS* point mutations, signaling events captured by this analysis may represent a heterogeneous mixture of networks. The possibility that specific *KRAS* mutants can present distinct shifts in signaling under the presence of WT protein should be further investigated. Nevertheless, our findings strongly encourage the notion that different genotypes, specifically the presence of the *KRAS* WT allele in *KRAS* mutant lung adenocarcinomas, prominently affect signaling events and response to treatment. Understanding the underlying mechanisms of these changes can potentially uncover new druggable targets and fulfill the unmet therapeutic needs of patients with *KRAS* mutant lung adenocarcinoma.

## Figures and Tables

**Figure 1 genes-12-01402-f001:**
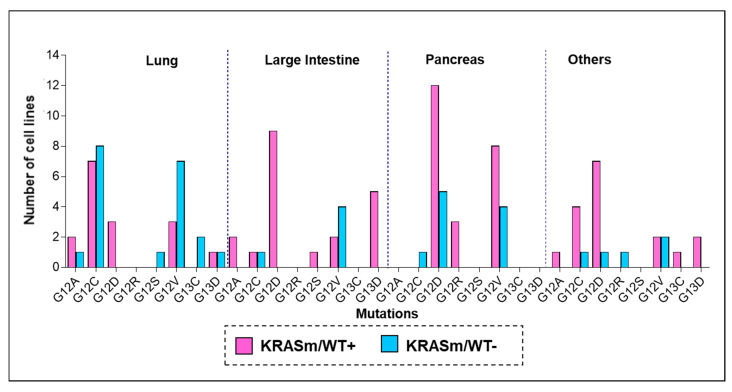
Frequency of *KRAS* oncogenic mutations across cancer cells lines of different origin. To explore frequency and distribution of *KRASm/WT+* and *KRASm/WT−* mutations across tumor types, *KRAS* zygosity was retrieved from the COSMIC and the NCI-funded KRAS initiative databases. Mutant *KRAS* cell lines were classified based on the presence (*KRASm/WT+* pink) or absence (*KRASm/WT− blue)* of the *KRAS* WT allele. Frequencies of *KRASm/WT+* and *KRASm/WT−* cell lines are displayed as bar graphs; mutations are listed on the x-axis, and number of cell lines identified for each variant is reported on the y-axis. Of the 116 identified cell lines, 38 were derived from lung lesions (32.7%), 33 from pancreatic cancers (28.4%), and 25 from tumors of the large intestine (21.5%). The 36 lung cancer models included in the analysis were established from the following tumors: 22 adenocarcinomas, 4 large cell carcinomas, 2 small cell lung cancers, 2 carcinomas not otherwise specified, 2 giant cell carcinoma, 1 adeno-squamous, 1 squamous carcinoma, 1 carcinoid and 1 epidermoid tumor.

**Figure 2 genes-12-01402-f002:**
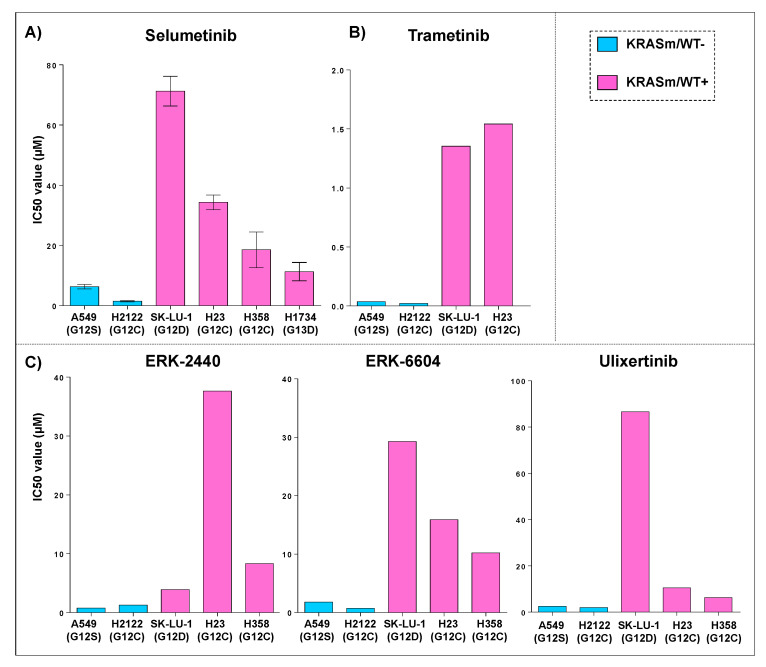
Sensitivity to MEK and ERK inhibitors in *KRASm/WT−* and *KRASm/WT+* adenocarcinoma cell lines. IC50 values for *KRAS* mutant cell lines treated with a kinase inhibitor targeting KRAS downstream substrates are displayed as bar graphs where cell lines are color-coded based on the presence *(KRASm/WT+* pink) or absence (*KRASm/WT−* blue) of the *KRAS* WT allele. IC50 average values (*n* = 4) and standard error of the mean after incubation with the MEK inhibitor Selumetinib for 72 h are displayed (**A**); models harboring *KRASm/WT−* mutations are more sensitive to MEK inhibition compared to cell lines retaining the wild-type copy of the *KRAS* allele (**A**). These trends were confirmed using data retrieved from the Genomics of Drug Sensitivity in Cancer (GDSC) database for the MEK inhibitor Trametinib (**B**) and the ERK inhibitors Ulixertinib, ERK6604 and ERK2440 (**C**). Single IC50 values are available for each compound on the GDSC database.

**Figure 3 genes-12-01402-f003:**
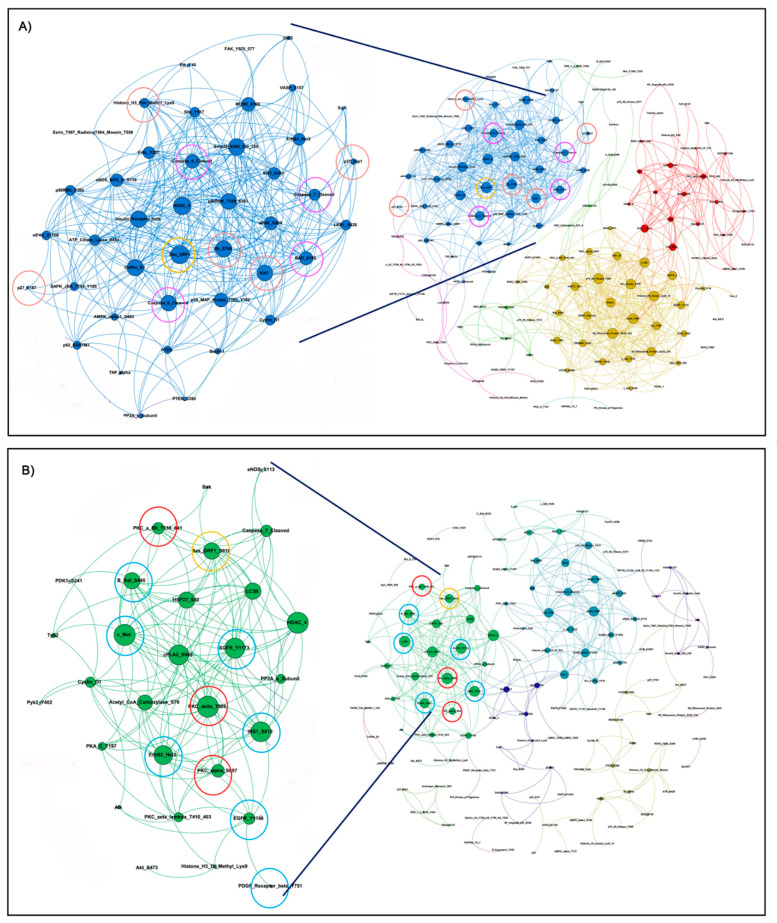
Interaction networks of signal transduction molecules in *KRASm/WT−* and *KRASm/WT+* NSCLC models. Spearman rank-order correlation coefficients of RPPA-based continuous values greater than 0.90 across the 183 measured analytes are displayed using network maps. Protein networks of *KRASm/WT−* (A549, H2122) show high levels of interaction with most proteins contained within three main clusters (**A**). Cell cycle regulators and proteins belonging to the apoptotic pathway are highlighted with red and pink circles, respectively. Network of *KRASm/WT+* cell lines (H1734, H23, H358) shows fewer interconnections compared to the *KRASm/WT−* cells (**B**). Receptor tyrosine kinase, MAPK signaling molecules and members of the PKC family are highlighted in red and green, respectively. Ras-GFR-1 is highlighted in yellow in both maps.

**Figure 4 genes-12-01402-f004:**
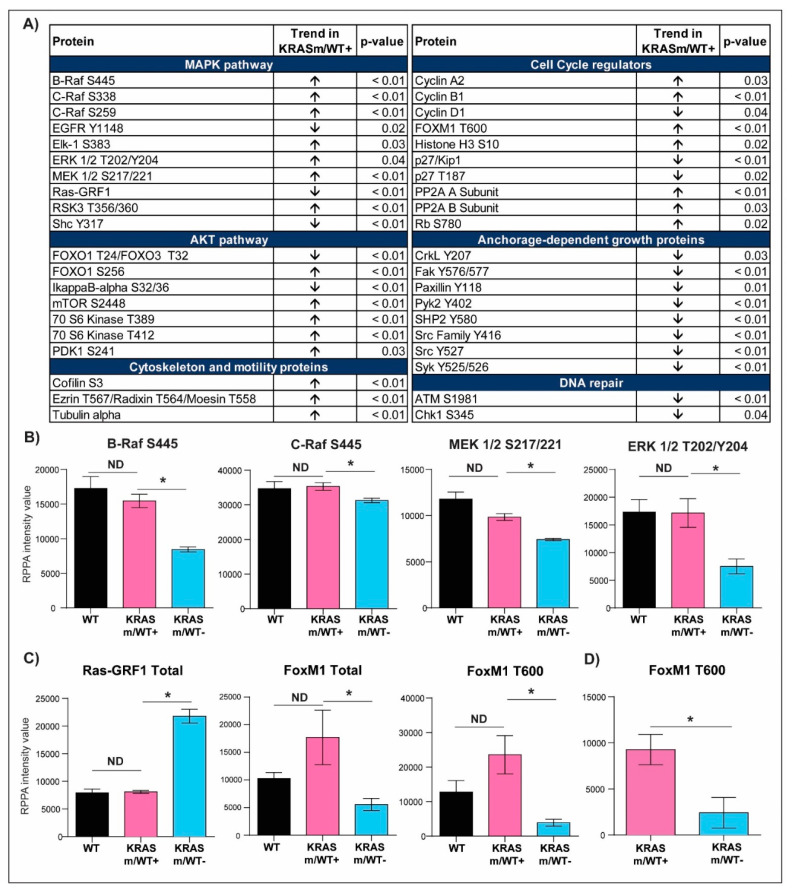
Selected signal transduction molecules differentially activated in *KRASm/WT+* and *KRASm/WT−* NSCLC models. Of the 183 signaling molecules measured by RPPA, 81 reached statistical significance when *KRASm/WT−* and *KRASm/WT+* cell lines were compared. Proteins belonging to the same signaling pathway were grouped based on their biological function and are displayed in (**A**). Arrows reflect trends in the *KRASm/WT+* cells (H1734, H23, H358) compared to *KRASm/WT−* (A549, H2122) models. Bar graphs displaying mean and standard error of the mean for member of the MAPK pathway are shown in (**B**). Of interest, while the activation of KRAS downstream signaling substrates reached statistical significance when *KRASm/WT−* and *KRASm/WT+* cell lines were compared (*), these differences were lost between *KRASm/WT+* and *KRAS* wild-type models (ND). Similar trends were also detected for Ras-GRF1, a modulator of RAS activity, and the cell cycle regulator FoxM1 (**C**). Differences in the activation of the cell cycle regulator FoxM1 between *KRASm/WT−* and *KRASm/WT+* tumors were confirmed in surgical specimens, suggesting clinical relevance for this finding (**D**). * Indicates comparisons that were statistically different (*p* < 0.05).

**Figure 5 genes-12-01402-f005:**
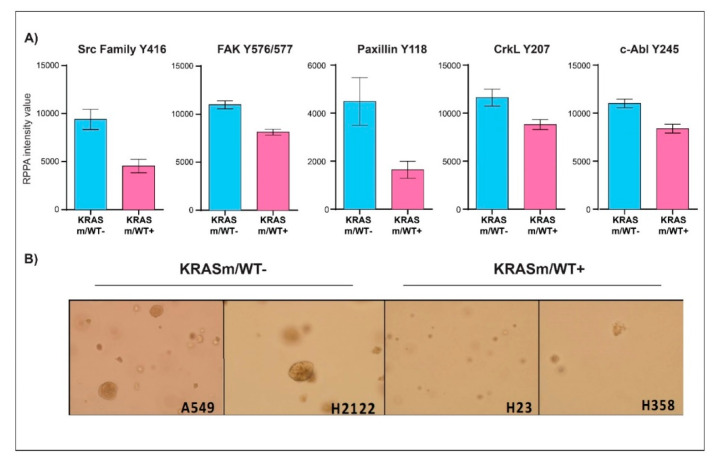
Anchorage-independent growth is enhanced in *KRASm/WT−* NSCLC cell lines. Bar graphs display mean and standard error of the mean of expression and activation of key signaling molecules involved in anchorage-independent growth in *KRASm/WT−* (A549, H2122) and *KRASm/WT+* (H1734, H23, H358) (**A**). Colony formation assay shows larger and more abundant colonies in the *KRASm/WT−* cell lines compared to *KRASm/WT+* models (**B**), confirming increased anchorage-independent growth in the *KRASm/WT* lines.

**Figure 6 genes-12-01402-f006:**
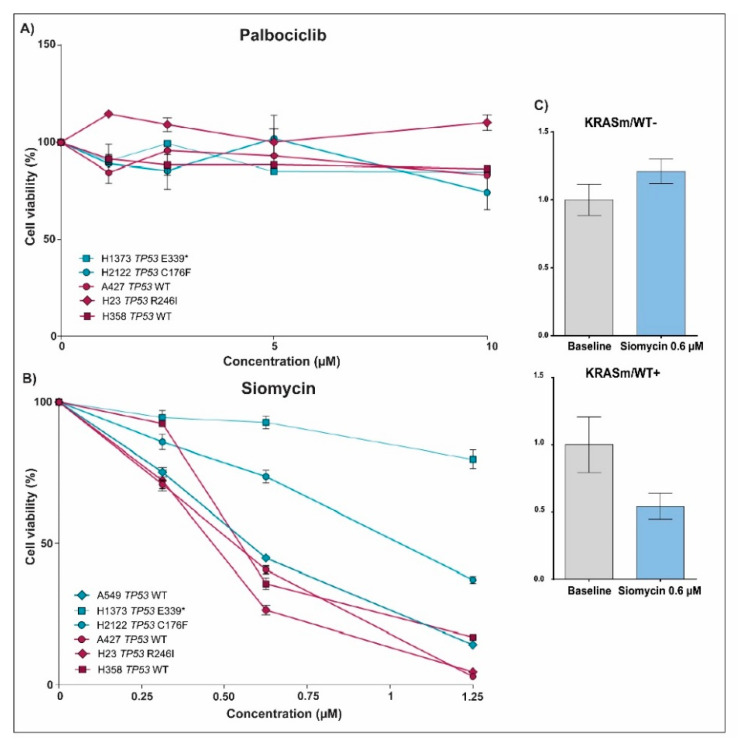
Drug sensitivity assay and changes in FoxM1 expression after treatment with Palbociclib and Siomycin in *KRASm/WT+* and *KRASm/WT−* NSCLC cell lines. Line plots show cell viability after 72 hours of incubation with Palbociclib and Siomycin in *KRASm/WT−* (A549, H1373 and H2122; blue lines) and *KRASm/WT+* (A427, H23 and H358; red lines) cell lines (**A**,**B**). Data were normalized on matched vehicle control samples, namely PBS and DMSO for Palbociclib and Siomycin, respectively. Changes in FoxM1 phosphorylation levels after 72 hours of incubation with 0.6 µM of Siomycin were captured in *KRASm/WT+* and *KRASm/WT−* lines using the RPPA (**C**). * Indicates stop codon.

**Figure 7 genes-12-01402-f007:**
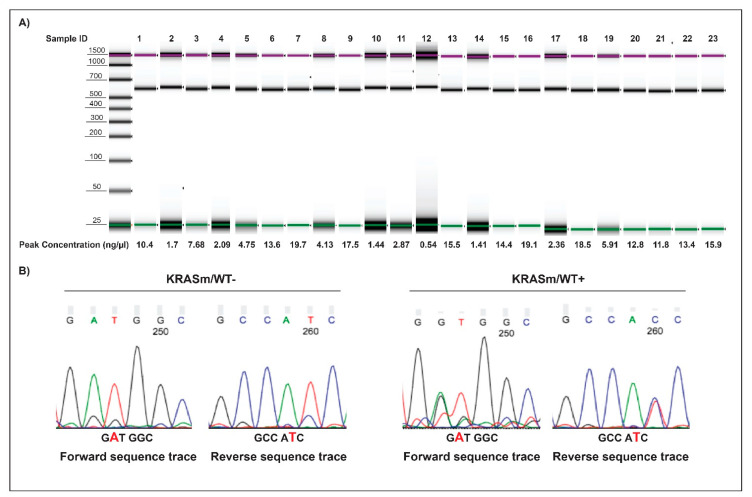
Amplicons and electropherogram of 23 microdissected NSCLC surgical specimens. Amplicons along with DNA concentration of the 23 microdissected biospecimens analyzed by PCR are displayed in (**A**). Examples of sequencing electropherograms with forward and reverse sequence of *KRASm/WT+* and *KRASm/WT−* samples harboring a *KRAS* G12D mutation are shown in (**B**). Samples were classified as *KRASm/WT−* when a single peak was detected at the mutation site.

## Data Availability

Data will be shared with interested researchers upon request.

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
