# Peer review of "Wild-Type KRAS Allele Effects on Druggable Targets in KRAS Mutant Lung Adenocarcinomas"

_genes, 2021, doi:10.3390/genes12091402_

Round 1

Reviewer 1 Report

The manuscript article "titled- Wild-type KRAS allele effects on druggable targets in KRAS mutant Non-Small Cell Lung Cancers" submitted to journal Genes by authors Baldelli et. al. demonstrates an interesting concept. The article is well written, easy to understand and flows well. The authors have also acknowledged the weakness of the article in the discussion section. However, I feel the manuscript will improve if the authors can add "how this study can be utilized clinically or in the future for patient treatment". Also, all legends need more details.

Author Response

Reviewer 1

The manuscript article "titled- Wild-type KRAS allele effects on druggable targets in KRAS mutant Non-Small Cell Lung Cancers" submitted to journal Genes by authors Baldelli et. al. demonstrates an interesting concept. The article is well written, easy to understand and flows well. The authors have also acknowledged the weakness of the article in the discussion section.

  1. However, I feel the manuscript will improve if the authors can add "how this study can be utilized clinically or in the future for patient treatment".

We thank the reviewer for the encouraging comments and suggestions. As recommended, we have added the following paragraph to the discussions addressing the clinical potential of these observations.

“Although the identification of druggable genomic alternations has revolutionized cancer treatment and management, predicting response to treatment still remains an unmet need in oncology. Zygosity and the presence of a wild-type copy of a mutant oncogene in particular, are rarely accounted for in the therapeutic decision-making process for cancer patients. However, emerging data suggest that KRAS zygosity may profoundly affect cancers’ phenotypes and responses to targeted compounds. For example, a recent paper by Liu and colleagues suggested that loss of the WT copy of the KRAS allele is not a rare event in lung adenocarcinomas and is associated with shorter survival [39]. In addition, this work confirms the unique role of zygosity as an important modulator of response to treatment in patients affected by KRAS mutant lung adenocarcinomas. These observations have important clinical implications as they suggest that KRAS zygosity may represent an underestimated pathological marker of KRAS mutant lung cancers. Understanding roles and effects of zygosity on signal transduction networks may open new opportunities for allocating patients to targeted treatments, understating mechanisms of resistance, and devising new single agent and combination treatments.”

  1. Also, all legends need more details.

We thank the reviewer for this suggestion. We have reworded the figure legends as described below:

Figure 1. Frequency of KRAS oncogenic mutations across cancer cells lines of different origin. To explore frequency and distribution of KRASm/WT+ and KRASm/WT- mutations across tumor types, KRAS zygosity was retrieved from the COSMIC and the NCI-funded KRAS initiative databases. Mutant KRAS cell lines were classified based on the presence (KRASm/WT+ pink) or absence (KRASm/WT- blue) of the KRAS WT allele and their frequencies are displayed as bar graphs. Mutations are listed on the x-axis and number of cell lines identified for each variant is reported on the y-axis. Of the 116 identified cell lines, 38 were derived from lung lesions (32.7%), 33 from pancreatic cancers (28.4%), and 25 from tumors of the large intestine (21.5%). The 36 lung cancer models included in the analysis were established from the following tumors: 22 adenocarcinomas, 4 large cell carcinomas, 2 small cell lung cancers, 2 carcinomas not otherwise specified, 2 giant cell carcinoma, and 1 adeno-squamous, 1 squamous carcinoma, 1 carcinoid, 1 epidermoid tumor.

Figure 2. Sensitivity to MEK and ERK inhibitors in KRASm/WT- and KRASm/WT+ adenocarcinoma cell lines. IC50 values for KRAS mutant cell lines treated with a kinase inhibitor targeting KRAS downstream substrates are displayed as bar graphs where cell lines are color-coded based on the presence (KRASm/WT+ pink) or absence (KRASm/WT- blue) of the KRAS WT allele. Panel A shows IC50 average values (n=4) and standard error of the mean after incubation with the MEK inhibitor Selumetinib for 72 hours. IC50 values suggest that models harboring KRASm/WT- mutations are more sensitive to MEK inhibition compared to cell lines retaining the wild-type copy of the KRAS allele (Panel A). These trends were confirmed using data retrieved from the Genomics of Drug Sensitivity in Cancer (GDSC) database for the MEK inhibitor Trametinib (Panel B) and the ERK inhibitors Ulixertinib, ERK6604, ERK2440 (Panel C). Single IC50 values are available for each compound on the GDSC database.

Figure 3. Interaction networks of signal transduction molecules in KRASm/WT- and KRASm/WT+ NSCLC models. Spearman rank-order correlation coefficients of RPPA-based continuous values greater than 0.90 across the 183 measured analytes are displayed using network maps. Protein networks of KRASm/WT- (A549, H2122) show high levels of interaction with most proteins contained within three main clusters (Panel A). Cell cycle regulators and proteins belonging to the apoptotic pathway are highlighted with red and pink circles, respectively. Network of KRASm/WT+ cell lines (H1734, H23, H358) shows less interconnections compared to the KRASm/WT- cells (Panel B). Receptor tyrosine kinase, MAPK signaling molecules, and members of the PKC family are highlighted in red and green, respectively. Ras-GFR-1 is highlighted in yellow in both maps.

Figure 4. Selected signal transduction molecules differentially activated in KRASm/WT+ and KRASm/WT- NSCLC models. Of the 183 signaling molecules measured by RPPA, 81 reached statistical significance when KRASm/WT- and KRASm/WT+ cell lines were compared. Proteins belonging to the same signaling pathway were grouped based on their biological function and displayed in Panel A. Arrows reflect trends in the KRASm/WT+ cells (H1734, H23, H358) compared to KRASm/WT- (A549, H2122) models.  Bar graphs displaying mean and standard error of the mean for member of the MAPK pathway are shown in Panel B. Of interest, while the activation of KRAS downstream signaling substrates reached statistical significance when KRASm/WT- and KRASm/WT+ cell lines were compared, these differences were lost between KRASm/WT+ and KRAS wild-type models. Similar trends were also detected for Ras-GRF1, a modulator of RAS activity, and the cell cycle regulator FoxM1 (Panel C). Differences in the activation of the cell cycle regulator FoxM1 between KRASm/WT- and KRASm/WT+ tumors were confirmed in surgical specimens, suggesting clinical relevance for this finding (Panel D).

Figure 5. Anchorage-independent growth is enhanced in KRASm/WT- NSCLC cell lines. Bar graphs display mean and standard error of the mean of the expression and activation of key signaling molecules involved in anchorage-independent growth in KRASm/WT+ (H1734, H23, H358) and KRASm/WT- (A549, H2122) (Panel A). Colony formation assay shows larger and more abundant colonies in the KRASm/WT- cell lines compared to KRASm/WT+ models (Panel B), confirming increased anchorage-independent growth in the KRASm/WT cell lines.

Figure 6. Drug sensitivity assay and changes in FoxM1 expression after treatment with Palbociclib and Siomycin in KRASm/WT+ and KRASm/WT- NSCLC cell lines. Line plots show cell viability after 72 hours of incubation with Palbociclib and Siomycin in KRASm/WT- (A549, H1373, and H2122; blue lines) and KRASm/WT+ (A427, H23, and H358; red lines) cell lines (Panel A). Data were normalized on matched vehicle control samples, namely PBS and DMSO for Palbociclib and Siomycin, respectively. Changes in FoxM1 phosphorylation levels after 72 hours of incubation with 0.6 µM of Siomycin were captured in KRASm/WT+ and KRASm/WT- lines using the RPPA (Panel B).

Figure 7. Amplicons and electropherogram of 23 microdissected NSCLC surgical specimens. Amplicons along with DNA concentration of the 23 microdissected biospecimens analyzed by PCR are displayed in Panel A. Examples of sequencing electropherograms with forward and reverse sequence of KRASm/WT+ and KRASm/WT- samples harboring a KRAS G12D mutation are shown in Panel B. Samples were classified as KRASm/WT- when a single peak was detected at the mutation site.

Reviewer 2 Report

The authors provided a manuscript of decent quality, exploring a very interesting hypothesis with potentially high clinical value. The article is well-divided. The integrity of introduction is sufficient however some additional information should be included. Then, the experimental parts are sufficiently explained in the material and methods sections. The results are extensively described and this leads to some confusion and a more careful revision is recommended. In particular, the figures have different problem that the authors should seriously consider. The conclusions are fair and well explained in parallel with the study’s findings.

A very crucial point I want to highlight is the material origin. I have understood that all the materials used in the study are lung adenocarcinoma. If so, I strongly suggest to authors to modify their conclusions accordingly and avoid using the terms non-small cell lung cancer, which includes also other tumor types such as squamous cell lung cancer and large cell cancer with different genetic status and oncogenic behavior.

In conclusion, the article would be considered for potential publication only after the authors complete the major revisions suggested below and reviewer in charge rechecks the revised manuscript.

Abstract:

line 18, remove the ending -s from the Cancers.

line 20, please remove the symbol ; and connect the sentences

line 21, the KRAS in italics

line 22, consider replacing the “tumors” with “NSCLC”

line 24, please replace the words targetable events. It is not very clear. Consider using candidate targets or druggable targets

line 27, add the word KRAS next to WT allele

line 30, impacts on signaling…...

line 31, remove the ending -s from the NSCLCs

Introduction:

line 42, please modify the abbreviation for squamous cell lung cancer. You can try SQLC for example. The SCLC abbreviation is more appropriate for Small Cell Lung Cancer.

lines 40-44, you are speaking for lung adenocarcinoma and squamous while after you mention the category of NSCLC. Please introduce the subtypes of NSCLC more smoothly, starting like this: In lung cancer and especially NSCLC, KRAS mutations are very frequent genomic events. KRAS is mainly detected in lung adenocarcinoma, the main subtype of NSCLC ……… etc.

line 45: precision medicine has been introduced only in lung adenocarcinoma and not generally in NSCLC. Please elaborate more writing brief information about the example of EGFR sensitizing mutations and tyrosine kinase inhibitors etc.

line 53: met with limited success.

line 53: remove the ; and connect the sentences.

line 59: use one of sotorasib or AMG510. Otherwise mention both also previously in line 57.

About KRAS inhibitors, write also about adagrasib, the other KRAS G12C inhibitor.

Line 70: both promote and…….

Line 80-83: last paragraph of introduction. Please rephrase it with more clear way. This way, it is difficult to understand.

Materials and methods:

All the cell lines are lung adenocarcinoma. Replace the NSCLC with lung adenocarcinoma.

Please add information about what type of inhibitors these compounds are (Selumetinib, Palbociclib etc.)

The tumor samples were lung adenocarcinoma?

Results:

Figure 1: remove the word “legend”.

You support that Cosmic data include lung cancer cell lines. However in the text you mention NSCLC while within the figure I see the word lung. So the Cosmic data include only NSCLC? Maybe these NSCLC are only lung adenocarcinoma cell lines?

Also I am not sure that NSCLC or lung cancer are more KRASm/WT+ compared to other tumor types. Large intestine looks the same. How the authors support this argument?

The figure will be improved if among the mutations there is a small gap.

line 254, please remind me how you determine that these cell lines have one wt allele.

When you mention the range, please put it from the lowest to the highest concentration.

Figure 2A, you mention KRAS wt cell line but I can not find it. Also, why there is the standard deviation only in the graph of selumetinib and not the rest of the inhibitors?

line 261, you mean inhibitors and not inhibitions?

line 295, KRAS is in italic but not the m(utant)

Figure 4, both groups are KRASmut/WT+. Supposed to be KRASmut/WT+ vs KRASmut/WT-. Which is which?

line 383, KRAS in italic but not the m(utant)

like 404, remove the words “in clinical biospecimens”

The legend of Figure 7 needs to be refined. What the panel 7B represents?

line 447, remove the “presented by Ambrogio and colleagues”

Please discuss the following paper which is line with authors rationale:

Loss of wild type KRAS in KRASMUT lung adenocarcinoma is associated with cancer mortality and confers sensitivity to FASN inhibitors

Author Response

Reviewer 2

The authors provided a manuscript of decent quality, exploring a very interesting hypothesis with potentially high clinical value. The article is well-divided. The integrity of introduction is sufficient however some additional information should be included. Then, the experimental parts are sufficiently explained in the material and methods sections. The results are extensively described and this leads to some confusion and a more careful revision is recommended. In particular, the figures have different problem that the authors should seriously consider. The conclusions are fair and well explained in parallel with the study’s findings.

We thank the reviewer for the encouraging comments and suggestions to improve the quality of our work. We have carefully analyzed and addressed the suggestions as indicated below.

  1. A very crucial point I want to highlight is the material origin. I have understood that all the materials used in the study are lung adenocarcinoma. If so, I strongly suggest to authors to modify their conclusions accordingly and avoid using the terms non-small cell lung cancer, which includes also other tumor types such as squamous cell lung cancer and large cell cancer with different genetic status and oncogenic behavior. In conclusion, the article would be considered for potential publication only after the authors complete the major revisions suggested below and reviewer in charge rechecks the revised manuscript.

We thank the reviewer for this comment. To accurately attribute our data to lung adenocarcinomas, we have changed the method and discussion sections as follow.

“A panel of eleven human adenocarcinoma cell lines including the A427, A549, Calu-3, H1373, H1734, H1838, H2122, H23, H358, H522 and SK-LU-1 models were obtained from the American Type Culture Collection (ATCC, Manassas, VA).”

The last paragraph of the discussion now reads as:

“Although the identification of druggable genomic alternations has revolutionized cancer treatment and management, predicting response to treatment still remains an unmet need in oncology. Zygosity and the presence of a wild-type copy of a mutant oncogene in particular, are rarely accounted for in the therapeutic decision-making process for cancer patients. However, emerging data suggest that KRAS zygosity may profoundly affect cancers’ phenotypes and responses to targeted compounds. For example, a recent paper by Liu and colleagues suggested that loss of the WT copy of the KRAS allele is not a rare event in lung adenocarcinomas and is associated with shorter survival [39]. In addition, this work confirms the unique role of zygosity as an important modulator of response to treatment in patients affected by KRAS mutant lung adenocarcinomas. These observations have important clinical implications as they suggest that KRAS zygosity may represent an underestimated pathological marker of KRAS mutant lung cancers. Understanding roles and effects of zygosity on signal transduction networks may open new opportunities for allocating patients to targeted treatments, understating mechanisms of resistance, and devising new single agent and combination treatments.”

  1. Abstract:

o             line 18, remove the ending -s from the Cancers.

o             line 20, please remove the symbol; and connect the sentences

o             line 21, the KRAS in italics

o             line 22, consider replacing the “tumors” with “NSCLC”

o             line 24, please replace the words targetable events. It is not very clear. Consider using candidate targets or druggable targets

o             line 27, add the word KRAS next to WT allele

o             line 31, remove the ending -s from the NSCLCs

o             line 30, impacts on signaling…...

We thank the reviewer for these suggestions, the suggested changes have been implemented in the new version of the manuscript.

  1. Introduction:

line 42, please modify the abbreviation for squamous cell lung cancer. You can try SQLC for example. The SCLC abbreviation is more appropriate for Small Cell Lung Cancer.

We apologize for the confusion created by the abbreviation. As requested by the reviewer, we have removed the SCLC abbreviation and have used the full squamous cell lung cancer name in the introduction.

  1. lines 40-44, you are speaking for lung adenocarcinoma and squamous while after you mention the category of NSCLC. Please introduce the subtypes of NSCLC more smoothly, starting like this: In lung cancer and especially NSCLC, KRAS mutations are very frequent genomic events. KRAS is mainly detected in lung adenocarcinoma, the main subtype of NSCLC ……… etc.

We thank the reviewer for this comment. We have rephrased the sentence in the introduction and it now reads:

“KRAS mutations are frequent genomic events in lung cancer, especially in NSCLC. Within NSCLC, KRAS mutations are almost exclusively seen in adenocarcinomas, with up to 40% of adenocarcinoma harboring oncogenic mutations of the KRAS gene. These mutations are rarely detected in the other lung cancer subtypes. Across the forty-four oncogenic KRAS mutations identified, 90% occur at codon 12, with G12C (40%) and G12V (22%) mutations being the most frequently observed in lung adenocarcinomas collectively [2–4].”

  1. line 45: precision medicine has been introduced only in lung adenocarcinoma and not generally in NSCLC. Please elaborate more writing brief information about the example of EGFR sensitizing mutations and tyrosine kinase inhibitors etc.

We thank the reviewer for the comment. We have included the reviewer suggestions in the introduction as follows:

“Undoubtedly, the introduction of precision medicine has revolutionized treatment for lung cancer patients. For example, small kinase inhibitors able to modulate the activity of aberrantly activated Epidermal Growth Factor Receptors (EGFR) are routinely used as first-line treatment options for EGFR mutant lung adenocarcinomas [5]. These targeted molecules have significantly improved response to treatment and survival for these patients. However, targeting adenocarcinomas harboring mutations of the EGFR downstream substrate KRAS still remains a significant unmet clinical need in oncology [5].”

  1. line 53: met with limited success.

This point has been addressed in the text.

  1. line 53: remove the ; and connect the sentences.

This point has been addressed in the text.

  1. line 59: use one of sotorasib or AMG510. Otherwise mention both also previously in line 57.

We thank the reviewer for this comment. We have included both Sotorasib and AMG510 in the text as suggested.

  1. About KRAS inhibitors, write also about adagrasib, the other KRAS G12C inhibitor.

According to the ongoing CodeBreak 100 study, treatment with the small G12C inhibitor Sotorasib (or AMG510) in combination with standard chemotherapy has shown objective remission in 33% of enrolled patients. Given these encouraging results, the FDA granted AMG510 approval as a treatment option for locally advanced or metastatic NSCLCs with G12C KRAS mutations. Another promising G12C inhibitor, Adagrasib, is also showing clinical activity in patients with advanced and previously treated NSCLC. According to the KRYSTAL-1 phase 1/2 trial results, in patients with advanced NSCLC G12C KRAS mutant tumors previously treated with chemotherapy and anti-PDL1 therapy, Adagrasib yielded 96% disease control rate with 45% objective response rate [13]. However, despite such promising data, maximizing the effect of these inhibitors as a monotherapy and understanding mechanisms of acquired resistance are still under investigation [14–16].”

  1. Line 70: both promote and…….

We have changed this sentence to “previous research has indicated that the WT KRAS protein can either promote or inhibit tumor progression in KRAS mutant tumors in a context dependent manner”.

  1. Line 80-83: last paragraph of introduction. Please rephrase it with more clear way. This way, it is difficult to understand.

The last paragraph of the introduction has been changed as follow: “Such approach provided us the unique opportunity to capture the effects of KRAS homo- and hetero-zygous mutations on druggable signal transduction molecules. Exploring the impact different genotypes can have on cell signaling events will help identify distinct druggable vulnerabilities and potentially advance precision medicine for KRAS mutant NSCLC patients.

Materials and methods:

  1. All the cell lines are lung adenocarcinoma. Replace the NSCLC with lung adenocarcinoma.

We thank the reviewer for this comment. We have addressed this issue in the method section that now reads as follow:

“A panel of eleven human adenocarcinoma cell lines including the A427, A549, Calu-3, H1373, H1734, H1838, H2122, H23, H358, H522 and SK-LU-1 models were obtained from the American Type Culture Collection (ATCC, Manassas, VA).”

We have also addressed this issue in the last paragraph of the discussion.

“While our data support the hypothesis that retention of the WT copy of the KRAS gene in mutant lung adenocarcinomas modulates signaling events and response to treatment, a few study limitations must be addressed. First, our observations are merely descriptive and based on a limited number of cell lines generated from lung adenocarcinomas. Given the highly variable genetic background of these models and different degree of dependency on mutant KRAS, generalizing our observations is premature and needs to be validated on larger datasets. Second, because we used cell lines harboring diverse KRAS point mutations, signaling events captured by this analysis may represent a heterogeneous mixture of networks. The possibility that specific KRAS mutants can present distinct shifts in signaling under the presence of WT protein should be further investigated. Nevertheless, our findings strongly encourage the notion that different genotypes, specifically presence of the KRAS WT allele in KRAS mutant lung adenocarcinomas, prominently affect signaling events and response to treatment. Understanding the underlying mechanisms of these changes can potentially uncover new druggable targets and fulfill the unmet therapeutic needs of patients with KRAS mutant lung adenocarcinomas.”

We have also changed the title to:

Wild-type KRAS allele effects on druggable targets in KRAS mutant Lung Adenocarcinomas”

  1. Please add information about what type of inhibitors these compounds are (Selumetinib, Palbociclib etc.)

To address this issue, we have modified the text as follow.

“In brief, for each cell line, a cell suspension containing 5000 to 7500 viable cells was seeded in 96-well plates 24h before treatment with either the small kinase MEK inhibitor Selumetinib (AZD6244, Selleckchem, Houston, TX), the Cdk 4/6 inhibitor Palbociclib (Selleckchem, Houston, TX), or the thiazole antibiotic known to modulate FoxM1 expression and transcriptional activity Siomycin (Cayman Chemical, Ann Arbor, MI).”

  1. The tumor samples were lung adenocarcinoma?

We thank the reviewer for this excellent question. Surgical biospecimens were collected from adenocarcinoma samples. We have added this information to the method section of the manuscript.

A total of 23 retrospective biospecimens from surgically treated KRAS mutant/EGFR WT lung adenocarcinomas collected between 2009 to 2013 at the S. Maria della Misericordia Hospital (Perugia, Italy) were analyzed.”

Results:

  1. Figure 1: remove the word “legend”.

We thank the reviewer for the suggestion. We have included in the manuscript a new updated figure where the word legend was removed.

  1. You support that Cosmic data include lung cancer cell lines. However, in the text you mention NSCLC while within the figure I see the word lung. So the Cosmic data include only NSCLC? Maybe these NSCLC are only lung adenocarcinoma cell lines?

We thank the reviewer for this very important observation. The 36 KRAS mutant lung cancer cell lines profiled by the COSMIC initiative were established from the following tumors: 22 lung adenocarcinomas, 4 large cell carcinomas, 2 small cell lung cancers, 2 carcinomas not otherwise specified, 2 giant cell carcinoma, and 1 adeno-squamous, 1 squamous carcinoma, 1 carcinoid, 1 epidermoid tumor.

We have added this information in the figure legend and have changed the text as follow:

Figure 1. Frequency of KRAS oncogenic mutations across cancer cells lines of different origin. To explore frequency and distribution of KRASm/WT+ and KRASm/WT- mutations across tumor types, KRAS zygosity was retrieved from the COSMIC and the NCI-funded KRAS initiative databases. Mutant KRAS cell lines were classified based on the presence (KRASm/WT+ pink) or absence (KRASm/WT- blue) of the KRAS WT allele and their frequencies are displayed as bar graphs. Mutations are listed on the x-axis and number of cell lines identified for each variant is reported on the y-axis. Of the 116 identified cell lines, 38 were derived from lung lesions (32.7%), 33 from pancreatic cancers (28.4%), and 25 from tumors of the large intestine (21.5%). The 36 lung cancer models included in the analysis were established from the following tumors: 22 adenocarcinomas, 4 large cell carcinomas, 2 small cell lung cancers, 2 carcinomas not otherwise specified, 2 giant cell carcinoma, and 1 adeno-squamous, 1 squamous carcinoma, 1 carcinoid, 1 epidermoid tumor.

  1. Also I am not sure that NSCLC or lung cancer are more KRASm/WT+ compared to other tumor types. Large intestine looks the same. How the authors support this argument?

We thank the reviewer for this comment. We realized that the sentence in the results missed the word “lacking”. We now have the following description in the text. “KRAS mutant tumors lacking the WT allele (KRASm/WT-) were more prevalent in lung cancer cell lines compared to other tumor types. Indeed, of the 36 lung cancer models retrieved form the COSMIC database, 47.2% were KRASm/WT-, compared to 30.3% in the pancreatic group and 20% in the large intestine lines.

  1. The figure will be improved if among the mutations there is a small gap.

We thank the reviewer for this suggestion. We have added gaps between the different mutations in figure 1.

  1. line 254, please remind me how you determine that these cell lines have one wt allele.

We thank the reviewer for this comment. As indicated in the method section “Presence of a wild-type copy of the allele for the following cell lines A427, H1734, H23, H358, SK-LU-1, Calu-3, H1838 and H522 were retrieved from the COSMIC database and confirmed by PCR.”

We have added the following text to former line 254 (now 382). “Response to Selumetinib was initially assessed on six commercially available KRAS mutant cell lines, of which four (H23, H358, H1734, and SK-LU-1) retained a copy of the WT allele. Presence of the WT allele in these models was first retrieved from the COSMIC database and subsequently confirmed by PCR analysis.”

  1. When you mention the range, please put it from the lowest to the highest concentration.

We thank the reviewer for the comment. We have updated the compounds’ ranges and they now read from the lowest to the highest concentration.

  1. Figure 2A, you mention KRAS wt cell line but I cannot find it. Also, why there is the standard deviation only in the graph of selumetinib and not the rest of the inhibitors?

We thank the reviewer for this comment. We reported IC50 for the WT cell lines in the text only. Because the range of the IC50 was very broad for the KRAS WT lines, adding them to the graph would have compressed the remaining data and rendered the KRASm/WT- data difficult to see.

Regarding the data presented in Panel B and C, these IC50 were retrieved from the GDSC database where a single IC50 value is provided for each compound. We have clarified this issue in the figure legend.

Figure 2. Sensitivity to MEK and ERK inhibitors in KRASm/WT- and KRASm/WT+ adenocarcinoma cell lines. IC50 values for KRAS mutant cell lines treated with a kinase inhibitor targeting KRAS downstream substrates are displayed as bar graphs where cell lines are color-coded based on the presence (KRASm/WT+ pink) or absence (KRASm/WT- blue) of the KRAS WT allele. Panel A shows IC50 average values (n=4) and standard error of the mean after incubation with the MEK inhibitor Selumetinib for 72 hours. IC50 values suggest that models harboring KRASm/WT- mutations are more sensitive to MEK inhibition compared to cell lines retaining the wild-type copy of the KRAS allele (Panel A). These trends were confirmed using data retrieved from the Genomics of Drug Sensitivity in Cancer (GDSC) database for the MEK inhibitor Trametinib (Panel B) and the ERK inhibitors Ulixertinib, ERK6604, ERK2440 (Panel C). Single IC50 values are available for each compound on the GDSC database.

  1. line 261, you mean inhibitors and not inhibitions?

We thank the reviewer for the comments. We have included the reviewer’s suggestion in the manuscript.

  1. Line 295, KRAS is in italic but not the m(utant)

We thank the reviewer for the comments. We have included the reviewer’s suggestion in the manuscript.

  1. Figure 4, both groups are KRASmut/WT+. Supposed to be KRASmut/WT+ vs KRASmut/WT-. Which is which?

We thank the reviewer for this comment. In figure 4 panel A, the column labeled KRASmut/WT+ refers to the trend of the measured analytes in KRASmut/WT+ models compared to the KRASmut/WT- cell line. For example, activation of B-RaF S445 was greater in the KRASmut/WT+ compared to KRASmut/WT- cell lines. We have added this explanation to the figure legend to further explain the nomenclature.

Figure 4. Selected signal transduction molecules differentially activated in KRASm/WT+ and KRASm/WT- NSCLC models. Of the 183 signaling molecules measured by RPPA, 81 reached statistical significance when KRASm/WT- and KRASm/WT+ cell lines were compared. Proteins belonging to the same signaling pathway were grouped based on their biological function and displayed in Panel A. Arrows reflect trends in the KRASm/WT+ cells (H1734, H23, H358) compared to KRASm/WT- (A549, H2122) models.  Bar graphs displaying mean and standard error of the mean for member of the MAPK pathway are shown in Panel B. Of interest, while the activation of KRAS downstream signaling substrates reached statistical significance when KRASm/WT- and KRASm/WT+ cell lines were compared, these differences were lost between KRASm/WT+ and KRAS wild-type models. Similar trends were also detected for Ras-GRF1, a modulator of RAS activity, and the cell cycle regulator FoxM1 (Panel C). Differences in the activation of the cell cycle regulator FoxM1 between KRASm/WT- and KRASm/WT+ tumors were confirmed in surgical specimens, suggesting clinical relevance for this finding (Panel D).

  1. line 383, KRAS in italic but not the m(utant)

We thank the reviewer for the comments. We have included the reviewer’s suggestion in the manuscript.

  1. like 404, remove the words “in clinical biospecimens”

We have updated the manuscript with the reviewer suggestion.

  1. The legend of Figure 7 needs to be refined. What the panel 7B represents?

We have updated the Figure 7 legend as suggested by the reviewer. Now it reads:

Figure 7. Amplicons and electropherogram of 23 microdissected NSCLC surgical specimens. Amplicons along with DNA concentration of the 23 microdissected biospecimens analyzed by PCR are displayed in Panel A. Examples of sequencing electropherograms with forward and reverse sequence of KRASm/WT+ and KRASm/WT- samples harboring a KRAS G12D mutation are shown in Panel B. Samples were classified as KRASm/WT- when a single peak was detected at the mutation site.

  1. line 447, remove the “presented by Ambrogio and colleagues”

The suggestion has been incorporated in the new version of the manuscript.

  1. Please discuss the following paper which is line with authors rationale:

“Loss of wild type KRAS in KRASMUT lung adenocarcinoma is associated with cancer mortality and confers sensitivity to FASN inhibitors”

“Although the identification of druggable genomic alternations has revolutionized cancer treatment and management, predicting response to treatment still remains an unmet need in oncology. Zygosity and the presence of a wild-type copy of a mutant oncogene in particular, are rarely accounted for in the therapeutic decision-making process for cancer patients. However, emerging data suggest that KRAS zygosity may profoundly affect cancers’ phenotypes and responses to targeted compounds. For example, a recent paper by Liu and colleagues suggested that loss of the WT copy of the KRAS allele is not a rare event in lung adenocarcinomas and is associated with shorter survival [39]. In addition, this work confirms the unique role of zygosity as an important modulator of response to treatment in patients affected by KRAS mutant lung adenocarcinomas. These observations have important clinical implications as they suggest that KRAS zygosity may represent an underestimated pathological marker of KRAS mutant lung cancers. Understanding roles and effects of zygosity on signal transduction networks may open new opportunities for allocating patients to targeted treatments, understating mechanisms of resistance, and devising new single agent and combination treatments.”

Round 2

Reviewer 2 Report

Dear authors,

You have done a great work with the revision of your manuscript. In my opinion, the article is now appropriate for publication.